# Occurrence of No-Harm Incidents and Adverse Events in Hospitalized Patients with Ischemic Stroke or TIA: A Cohort Study Using Trigger Tool Methodology

**DOI:** 10.3390/ijerph19052796

**Published:** 2022-02-27

**Authors:** Bartosch Nowak, René Schwendimann, Philippe Lyrer, Leo H. Bonati, Gian Marco De Marchis, Nils Peters, Franziska Zúñiga, Lili Saar, Maria Unbeck, Michael Simon

**Affiliations:** 1Department Head Organs, Spine- and Neuromedicine, University Hospital Basel, 4031 Basel, Switzerland; bartosch.nowak@usb.ch; 2Patient Safety Office, University Hospital Basel, 4031 Basel, Switzerland; rene.schwendimann@usb.ch; 3Institute of Nursing Science, University of Basel, 4031 Basel, Switzerland; franziska.zuniga@unibas.ch; 4Department of Neurology and Stroke Center, University Hospital and University of Basel, 4031 Basel, Switzerland; philippe.lyrer@usb.ch (P.L.); leo.bonati@usb.ch (L.H.B.); gian.demarchis@usb.ch (G.M.D.M.); nils.peters@usb.ch (N.P.); 5Department of Neurology, Universitätsklinik Freiburg, 79106 Freiburg im Breisgau, Germany; lili.maria.saar@uniklinik-freiburg.de; 6School of Health and Welfare, Dalarna University, 79131 Falun, Sweden; mun@du.se; 7Department of Neurobiology, Care Sciences and Society, Karolinska Institutet, 17177 Stockholm, Sweden

**Keywords:** adverse events, no-harm incidents, retrospective record review, stroke, trigger tool methodology

## Abstract

Adverse events (AEs)—healthcare caused events leading to patient harm or even death—are common in healthcare. Although it is a frequently investigated topic, systematic knowledge on this phenomenon in stroke patients is limited. To determine cumulative incidence of no-harm incidents and AEs, including their severity and preventability, a cohort study using trigger tool methodology for retrospective record review was designed. The study was carried out in a stroke center at a university hospital in the German speaking part of Switzerland. Electronic records from 150 randomly selected patient admissions for transient ischemic attack (TIA) or ischemic stroke, with or without acute recanalization therapy, were used. In total, 170 events (108 AEs and 62 no-harm incidents) were identified, affecting 83 patients (55.3%; 95% CI 47 to 63.4), corresponding to an event rate of 113 events/100 admissions or 142 events/1000 patient days. The three most frequent AEs were ischemic strokes (*n* = 12, 7.1%), urinary tract infections (*n* = 11, 6.5%) and phlebitis (*n* = 10, 5.9%). The most frequent no-harm incidents were medication events (*n* = 37, 21.8%). Preventability ranged from 12.5% for allergic reactions to 100% for medication events and pressure ulcers. Most of the events found (142; 83.5%; 95% CI 76.9 to 88.6) occurred throughout the whole stroke care. The remaining 28 events (16.5%; 95% CI 11.4 to 23.1) were detected during stroke care but were related to care outside the stroke pathway. Trigger tool methodology allows detection of AEs and no-harm incidents, showing a frequent occurrence of both event types in stroke and TIA patients. Further investigations into events’ relationships with organizational systems and processes will be needed, first to achieve a better understanding of these events’ underlying mechanisms and risk factors, then to determine efforts needed to improve patient safety.

## 1. Introduction

Adverse events (AEs) commonly cause patients temporary or permanent disability [1,2], or even death. Other consequences include extended hospital stay length [3,4] and increased healthcare costs [5]. AEs are caused by care providers and have no direct relationship to the patient’s underlying condition [6]. In general, roughly one patient in ten is affected by an AE during in-hospital treatment [1,2]. Although AEs are a frequently investigated topic in healthcare [1,2], little is known about their occurrence in stroke patients. While few studies have investigated this phenomenon in stroke services [3,7,8], those that have reported a wide range of AE incidence: from 2.8% [8] to 63% [3], with up to 47% [7] considered preventable. Infections [4], hospital-acquired thromboembolism, strokes during interventional procedures, falls [7] and AEs resulting from recombinant tissue plasminogen activator (rtPA) administration, such as bleeding, are among the most common. And among rtPA-related AEs, intra-cerebral hemorrhage is the most serious [8]. These studies, focusing on stroke patients, used different methods for event detection, such as Global Trigger Tool (GTT) methodology [8], voluntary and mandatory reporting systems [7] and retrospective record review [3].

Based on modern patient safety concepts, positive and negative outcomes can occur form the same system [9,10]. It is likely that no-harm incidents (those that reached the patient but caused no discernable harm) [11] have the same contributing factors as AEs. The main difference is that, in these cases, patients were not harmed. Putting an additional focus on no-harm incidents provides valuable information on areas for improvement and learning from how to anticipate and reduce emerging risks before patient harm occurs. Investigating AEs and no-harm incidents is important to gain knowledge and understanding of how to improve patient safety [12]. Therefore, the information on no-harm incidents provides an additional perspective on patient safety during care services and the opportunity to improve areas at risk for patient harm [13].

A well-known and established method for the detection of events is that of retrospective record reviews, using trigger tool methodology [13,14,15]. One commonly used tool is the Global Trigger Tool (GTT) [6]. Using routinely collected care data from the patient’s record, the GTT review process is carried out by a review team in two stages. The primary reviewers, who need to be familiar with both the local documentation structure and with the screened population’s clinical background, conduct the first review independently. This involves screening each record for predefined triggers, which may hint at potential events. If triggers are found, the record is searched in more detail for potential events. If an event is detected, a severity rating is performed, and a consensus reached between the primary reviewers. In a second review stage, the consensus is presented to a physician for result verification [6]. Compared to voluntary reporting systems, this stage’s main advantage is its substantially higher sensitivity, regarding event detection [14].

The GTT consists of six trigger modules (e.g., care module, medication module). In some studies, all of these modules are used [16,17], while most researchers use only those modules relevant to their clinical setting [18,19]. However, other users of the GTT methodology design additional modules (e.g., oncology module [20,21]) or create modified versions, tailored specifically to their clinical context [8,15,22]. While some versions diverge too much from the original GTT to label it as GTT, they are still considered as trigger tools (TTs). Even though TT methodology usually focuses on the detection of AEs [6], it is also possible to identify no-harm incidents during the same retrospective record review process [13,23,24]. We, therefore, assume that during stroke services, it is possible to identify both no-harm incidents and AEs, using TT methodology.

Working with a randomly selected sample of in-hospital patients treated for ischemic stroke or transitory ischemic attack (TIA), in a Swiss stroke center, this study’s purpose was to determine cumulative incidence of no-harm incidents and AEs, including their severity and preventability.

## 2. Materials and Methods

### 2.1. Study Design

We conducted a retrospective cohort study following TT methodology for the review of routinely collected patient data from electronic healthcare records (EHRs) [6].

### 2.2. Setting and Sample

The study was conducted in a certified stroke center integrated in a 770-bed university hospital in the German-speaking part of Switzerland. For the acute phase, patients usually stay 24–72 h in the stroke unit before being transferred to the neurological ward. The entire stoke response pathway, which includes the patient’s admission, in-hospital treatment, care and diagnostics regarding stroke, will henceforth be called stroke service. Our review included a screening of the full EHR data for each admitted patient. Due to similar pathogenic causes and treatment options, this study focused on patients with ischemic stroke or TIA. Patients who had provided general consent were aged 18 years or older, presented with TIA or ischemic stroke, who were admitted between 1 April 2017 and 31 March 2018 to the stroke center, with or without acute recanalization therapy and an in-hospital stay of at least 24 h, were eligible for this study. Patients with hemorrhagic stroke were excluded. During the study period, 1090 patients (hemorrhagic stroke *n* = 82, any kind of ischemic stroke *n* = 885, TIA *n* = 123) were admitted to the stroke center. For this study, 424 patients met the inclusion criteria and therefore were eligible for the random sampling. Due to resource constraints, a sample size of 150 EHRs was drawn by B.N. using R statistical software [25]. No formal power analysis was conducted for this study. Based on an AE rate of approximately 10% [1,2] and a sample size of *n* 150, a 95% confidence interval of 3.4–12.2 was expected.

### 2.3. Definitions

An AE is “unintended physical injury resulting from or contributed to by medical care that requires additional monitoring, treatment or hospitalization, or that results in death” [6]. No-harm incidents are events that reached the patient but caused no discernible harm [11]. A preventable event was defined as one that could have been prevented if adequate actions had been taken during the patient’s contact with healthcare [26].

### 2.4. Measurement and Variables

Events were detected using 36 triggers, i.e., specific hints that commonly indicate AEs. The triggers used were based on the GTT’s 15-trigger care module (e.g., fall) and its 13-trigger medication module (e.g., vitamin K administration) [6] and were enriched with eight self-developed stroke-specific triggers (e.g., endovascular treatment). These triggers were based on potential events in connection to stroke treatment and used stroke triggers from the literature [8] after discussion with the research team and senior stroke physicians. Our targets included events arising from acts both of commission (active care delivery) and of omission.

When a potential event was found via the triggers, a four-point Likert-type scale was used to determine whether or not it was care-related (1= not related to care, 2 = probably not related to care, 3 = probably related to care, 4 = related to care). If the event was judged as 3 or 4, the reviewer continued the review process. To gauge preventability, a similar four-point Likert-type scale was used (1 = not preventable, 2 = probably not preventable, 3 = probably preventable, 4 = preventable). This rating was dichotomized into either not preventable (1 and 2) or preventable (3 and 4) [27]. To determine severity, a modified version of the recommended National Coordinating Council for Medication Error Reporting and Prevention (NCC MERP) Index was applied [6,27,28]. This scale consists of nine categories lettered A–I. Categories A and B indicate respectively potential errors and errors that never reached the patient; C and D indicate no-harm incidents; category E–I are considered AEs. Events categorized as C to I were included in this study. To represent a broad patient perspective, we collected all events, independent of their origin.

Patient demographics were manually extracted from the EHR by B.N. Length of stay (LOS) and the modified Rankin Scale (mRS) data were retrieved from the stroke center’s data in the Swiss Stroke Registry [29]. The mRS is a seven-level scale that gauges functional outcomes after stroke (range: no symptoms–dead) [30,31]. National Institutes of Health Stroke Scale (NIHSS) ratings were extracted from routinely collected data during the review process.

### 2.5. Data Collection

From January 2019 to March 2019, two trained registered nurse reviewers with clinical backgrounds in stroke care and knowledge of the EHR structure (B.N. and L.S.) independently reviewed case records for each selected patient’s inpatient stay during the sample period (see Appendix A). As a first step, all nurse, physician and therapist EHR documentation related to the selected admissions was systematically checked for occurrences of the predefined triggers. To allow a thorough baseline review, the time for primary review was not limited. Each reviewer recorded their findings in a self-developed study-specific protocol. In case of positive triggers, the EHR was searched in more detail to detect information about the potential event. If a potential event was identified, the primary reviewer evaluated whether it was care-related. This step was adopted from another TT study [24]. If an event was care-related, its severity rating, preventability determination, and type were set. Both reviewers met regularly to discuss and find consensus on the collected data. In the second review stage, B.N. presented the findings from the screening protocol to a senior stroke physician to verify the results. In cases of uncertainty or open questions, the patient record review was repeated until a consensus was reached.

### 2.6. Data Analysis

Descriptive statistics, e.g., frequencies, events per 100 admissions and per 1000 patient days, mean, standard deviation (SD), t-test for continuous and chi-squared test for categorical variables and 95% CI were compiled and analyzed. For result calculation, statistical software R, version 3.5.1 [25] and Microsoft Excel, 365 ProPlus (version 2010) [32] were used.

The positive predictive value (PPV) was calculated for each trigger and for this study refers to the number of times triggers have led to a specific AE or no-harm incident, divided by the overall number of times this particular trigger was found [22]. An overview of all triggers can be found in the Appendix A.

## 3. Results

The study sample consisted of 150 inpatients (110 males (73.3%)); overall mean LOS 8 days (SD = 5.5). In 135 patients (90%), the reason for admission was a cerebral ischemic infarction. As acute treatment, 32 patients (21.3%) received systemic rtPA, EVT or a combination of both. The mean NIHSS score on admission was 3.8 (SD = 4.3). Patients affected by an AE (*n* = 64) stayed 3.5 days (*p* = <0.001) longer in the hospital, had a more severe stroke (mean NIHSS 5.8, SD=5.2, *p* = 0.009) and received acute revascularization therapy more often (*p* = 0.001). More details are displayed in Table 1.

### 3.1. Identified Events

We identified 170 events, of which 62 (36.5%) were no-harm incidents involving 43 patients; the remaining 108 (63.5%) were AEs, affecting 64 patients. In total, 83 (55.3%) patients were affected by both types of events, with an overall event occurrence of 113 events per 100 admissions, and 142 events per 1000 patient days, respectively.

Of the total events, 110 (64.7%) were judged as preventable. Of those, 51 were no-harm incidents, involving 36 patients; the remaining 59 were AEs, affecting 41 patients. Overall, 61 patients were affected by preventable events, with a preventable occurrence rate of 73 events per 100 patients and 93 events per 1000 patient days. More details are presented in Table 2.

### 3.2. Detailed Event Presentation

Of the 170 identified events, 46 (27.1%) were related to general care. The two most common events within this category were phlebitis (*n* = 10, 5.9%) and constipation lasting five days or longer (*n* = 7, 4.1%). We also detected 37 (21.8%) medication events, for example, wrong prescriptions, which were the most frequent no-harm incidents. Bleeding (various types) occurred eleven times (6.5%). These included three (1.8%) instances of epistaxis and two (1.2%) of gastrointestinal bleeding. Neurological events appeared 30 (17.6%) times, including 12 (7.1%) cases of ischemic stroke, of which three (1.8%) were ischemic re-strokes and nine (5.3%) were new strokes. These 12 strokes included small cortical infarctions without clinical impact, as well as severe strokes that led to permanent disability. We also identified three (1.8%) cases of intracerebral bleeding, as a complication of rtPA administration. Within the category of healthcare-associated infections, urinary tract infections were the most frequent AE, with eleven (6.5%) events. Of the eleven internal events, statin induced myalgia / myopathy accounted for three (1.8%). Of eight (4.7%) allergic reactions, rashes appeared in five (2.9%). Eight falls (4.7%) were identified, five (2.9%) without injury, three (1.8%) with local contusions. Four (2.4%) category I pressure ulcers were found (2.4%). An overview of the occurred event types can be found in Table 3. The full table is presented in the Appendix A, as well as data regarding interrater reliability, screening time and trigger outcomes.

### 3.3. Severity Rating and Preventability

Accounting for 36.5% (*n* = 52) of all events, no-harm incidents are represented on the NCC MERP index, in categories C and D. Temporary patient harm, with or without prolonged hospitalization or outpatient treatment (category E and F), was found in 54.1% (*n* = 92) of all events. In 8.8% (*n* = 15) of cases, patients suffered permanent harm due to ischemic stroke, brain parenchymal bleeding or epilepsies (category G). One patient suffered life-threatening gastrointestinal bleeding (category H). No patient deaths were found during the review process.

The preventability ranged broadly (12.5–100%) between event categories. For example, while all medication events and pressure ulcers were considered preventable, healthcare-associated infections and general care-related events were rated as preventable in 80% and 78.3% of the cases, respectively. Allergic reactions and internal events showed the lowest preventability rates: respectively, 12.5% and 27.3%.

Most of the found events (142, 83.5%) occurred during the stroke care of the initial stroke admission. The origin of these events was in direct relation to stroke care. The remaining 28 (16.5%) events were found during the review, but the origin of these events was related to other care services, outside the stroke care (Table 4).

### 3.4. Trigger Occurrence and Positive Predictive Value on Appended Stroke Triggers

The stroke-specific triggers were identified 106 times during primary review, leading to a total PPV of 14.2%. Fourteen times, four triggers indicated the occurrence of twelve AEs (parenchymal bleeding, epistaxis, other allergic reactions, new stroke and re-stroke, other cerebrovascular events and gingival bleeding). Three triggers were identified but were assessed not to be related to any no-harm incident or AE. One trigger was not identified (Table 5). A detailed presentation of all triggers can be found in the Appendix A.

## 4. Discussion

Using retrospective record reviews, following GTT methodology [6], we conducted a retrospective cohort study on no-harm incidents’ and AEs’ occurrence of 150 randomly selected patients, treated for ischemic stroke or TIA. We found 170 events relating to 83 patients (55.3%).

Very few studies report results on AE occurrence in stroke patients; among those that do, the variation in results is very wide [3,7,8]. Studies from stroke care settings with a similar focus range from 2.8–63% for AEs [3,7,8]. This reflects their wide range of clinical settings (e.g., emergency departments), patient samples (e.g., patients with cerebral bleeding), detection methods (e.g., voluntary reporting systems) and even definitions of AEs, all of which limit direct comparison of results. However, regardless of the setting, sample, or detection methods, one finding is consistent: care-related events that can or do harm stroke patients are common.

Preventable events were found among both AEs and no-harm incidents. Of 108 AEs detected, just over half (54.6%) were deemed preventable. Of the three most common preventable event categories, two—medication events and pressure ulcers (category I)—are both 100% preventable, while the third ranked category, healthcare-associated infections, are 80% preventable. Regarding stroke service-related no-harm incidents and AEs, we classed 65.5% of cases as preventable. This number is high compared to another university hospital with an integrated stroke service, which judged 47% of found events preventable [7]. Although preventability was determined via different methods, both sets of findings indicate a strong potential to adapt care systems to prevent patients from harm. Taking a range of care setting characteristics into account, one systematic review and meta-analysis identified a preventability rate of 55% [33]. The next step is to analyze each event type separately, focusing on its mechanisms of occurrence and contributing factors.

Our modified version of the TT allows a more inclusive review of events to maximize our data’s value. It includes the capacity to detect and evaluate acts, both of commission and of omission. In fact, our comprehensive review, theoretically, could detect virtually any possible event documented in the EHR. No time limit applied to the primary review and both no-harm incidents and AEs were collected. All events were classified according to the NCC MERP index classifications. Even if not specifically recommended by the GTT white paper [6], we additionally gauged each event’s preventability. These adaptions likely increased our review method’s sensitivity, explaining our higher-than-usual incidence rate. We also created a stroke-specific module to detect AEs, such as bleedings or any neurovascular events, due to stroke-specific treatments. However, adding eight stroke-specific triggers to the original TT likely didn’t increase the chance of higher event detection rate. Four stroke-specific triggers (systemic administration of rtPA, EVT, neurologic decline of the NIHSS ≥ 4 from the initial score and neurologic decline of the GCS ≥ 4 from the initial score) identified twelve AEs, such as parenchymal bleeding, epistaxis, other allergic reactions, new stroke and re-stroke, other cerebrovascular events, and gingival bleeding. All of these could have been identified by other triggers, such as any procedure complications or in-hospital stroke. A comparison of record reviews with two teams, one using the original GTT modules, the other using an additional oncology module, showed no significant differences on the rate of found events [20]. Even though we do not have a direct comparison, the low PPV of 14.2% of the total stroke appended triggers supports our assumption, that no additional value was made.

Customizing the GTT is common practice. In general, modifying the GTT includes the AE definition, the number of reviewers, sample size, harm severity ratings, including events related to omission, preventability judgment, and method of reporting AE rates [15]. The one previous study to use GTT methodology in stroke patients customized the tool to identify specific AEs within the first 24 h after rtPA administration. In that case, only 14 AEs were found in 498 patients. Rather than classifying harm severity for every AE using the NCC MERP index, only intra-cerebral bleedings received classification (via the intra-cerebral hemorrhage scale) [8,34]. One of the GTT’s strengths is that it can be customized to make it either more inclusive or more exclusive for event detection. To facilitate the provision of robust data, particularly for further investigations into issues affecting patient safety, we recommend tailoring the GTT as necessary to the targeted clinical setting. 

### Strengths and Limitations

For the first time, this study used TT methodology to detect no-harm incidents and AEs, including acts both of commission and of omission, in TIA and stroke patients [6]. Each record was double-reviewed with the consensus data, then subjected to analyses. Furthermore, any detected event was classified regarding its severity using the NCC MERP index, and its preventability determined. This baseline determination provides robust results for further investigation and international comparison. The collection of no-harm incidents and AEs identified risk areas for further improvement, to avoid or mitigate patient harm.

This first explorative pilot study also has certain limitations. Most notably, it was a single center study, using a relatively small sample of 150 admissions. Furthermore, we applied descriptive statistics to present epidemiological data and did not correct for specific patient characteristics and conditions that might increase the likelihood of event occurrence. In addition, our record review methodologies rely entirely on events documented in the records. Therefore, comprehensive event detection is dependent on proper and complete documentation. If iatrogenic events are not recorded, they cannot be detected [35].

## 5. Conclusions

Retrospective record reviews, using TT methodology, was perceived as a sensitive system of detecting both AEs and no-harm incidents during stroke service. Although the results may not be generalized to other stroke centers, they highlight the high frequency of events that affect patient safety in stroke care. Both to improve our understanding of such events’ underlying mechanisms and to support interventions and precautions, to tackle the related phenomena, we recommend further investigation into their relationships with organizational processes and structures.

## Figures and Tables

**Table 1 ijerph-19-02796-t001:** Demographics and clinical characteristics.

Demographic and Clinical Categories	Frequency	Patients with No AE *n* = 86	Patients with AE *n* = 64	*p*-Value
Age in years, mean (SD) ^1^	71.8 (13.3)	72.1 (12.5)	71.4 (14.5)	0.756
Sex				
Men, *n* (%)	110 (73.3)	70 (81.4)	40 (62.5)	0.016
Female, *n* (%)	40 (26.7)	16 (18.6)	24 (37.5)	
LOS ^2^				
Patient days, mean (SD)	8.0 (5.5)	6.5 (4.8)	10.0 (5.8)	<0.001
Patient days, total	1192			
Cerebrovascular events, *n* (%)				0.062
Ischemic stroke	135 (90)	85 (98.8)	64 (100)	
TIA ^3^	15 (10)	1 (1.2)	0 (0.0)	
Acute treatment, *n* (%)				0.001
Conservative	118 (78.7)	75 (87.2)	43 (67.2)	
Systemic rtPA ^4^	22 (14.7)	11 (12.8)	11 (17.2)	
EVT ^5^	6 (4.0)	0 (0.0)	6 (9.4)	
rtPA and EVT	4 (2.7)	0 (0.0)	4 (6.2)	
Clinical metrics				
NIHSS ^6^ on admission, mean (SD)	3.8 (4.3)	2.3 (2.4)	5.8 (5.2)	0.009
mRS ^7^ after 3 months, mean (SD)	1.3 (1.4)	1.1 (1.4)	1.6 (1.5)	0.201

Abbreviations: ^1^ standard deviation, ^2^ LOS length of stay, ^3^ TIA transitory ischemic attack, ^4^ rtPA recombinant tissue plasminogenactivator, ^5^ EVT endovascular treatment, ^6^ NIHSS National Institution of Health Stroke Scale (0-42), ^7^ mRS modified Rankin Scale (0-6).

**Table 2 ijerph-19-02796-t002:** Overview of no-harm incidents and AE occurrence.

	No-Harm Incidents	AEs ^1^	Total
Event overview			
No ^2^ of events	62	108	170
No of affected patients, *n* (%; 95% CI)	43 (28.7; 21.7–36.7)	64 (42.7; 34.7–51.0)	83 (55.3; 47–63.4)
No of patients with >1 event (%)	12 (8.0)	27 (18.0)	39 (26.0)
No of events per 100 admissions	41	72	113
No of events per 1000 patient days	52	90	142
Preventable events			
No of preventable events	51	59	110
No of affected patients, *n* (%; 95% CI)	36 (24.0; 17.6–31.8)	41 (27.3; 20.5–35.3)	61 (40.7; 32.8–49)
No of patients with >1 preventable event (%)	10 (6.7)	11 (7.3)	21 (14)
No of preventable events per 100 admissions	34	39	73
No of preventable events per 1000 patient days	43	50	93

Abbreviations: ^1^ AE adverse event, ^2^ No number.

**Table 3 ijerph-19-02796-t003:** Event types.

	No-Harm Incidents, *n* (%; 95% CI)	AEs ^1^, *n* (%; 95% CI)	Total, *n* (%; 95% CI)
General care related events	17 (10; 6.1–15.8)	29 (17.1; 11.9–23.7)	46 (27.1; 20.7–34.5)
Medication events	37 (21.8; 16–28.9)	-	37 (21.8; 16.0–28.9)
Neurologic events	1 (0.6; 0.03–3.7)	29 (17.1; 11.9–23.7)	30 (17.6; 12.4–24.4)
Healthcare-associated infections	-	15 (8.8; 5.2–14.4)	15 (8.8; 5.2–14.4)
Bleedings	-	11 (6.5; 3.4–11.6)	11 (6.5; 3.4–11.6)
Internal events	1 (0.6; 0.03–3.7)	10 (5.9; 3.0–10.8)	11 (6.5; 3.4–11.6)
Allergic reactions	-	8 (4.7; 2.2–9.4)	8 (4.7; 2.2–9.4)
Falls	6 (3.5; 1.4–7.9)	2 (1.1; 0.2–4.6)	8 (4.7; 2.2–9.4)
Pressure ulcers, category I	-	4 (2.4; 0.8–6.3)	4 (2.4; 0.8–6.3)
Total, *n* (%)	62 (36.5)	108 (63.5)	170 (100)

Abbreviations: ^1^ AE adverse event.

**Table 4 ijerph-19-02796-t004:** Severity rating, preventability, and relation to stroke service.

Event Category	Severity Rating NCC MERP ^1^ Index							Preventability	Stroke Service Relation
	C	D	E	F	G	H	I	Preventable Events	
	*n* (%)	*n* (%)	*n* (%)	*n* (%)	*n* (%)	*n* (%)	*n* (%)	*n* (%)	*n* (%)
General care-related events	9 (19.6)	8 (17.4)	26 (56.5)	3 (6.5)	-	-	-	36 (78.3)	38 (82.6)
Medication events	35 (94.6)	2 (5.4)	-	-	-	-	-	37 (100)	35 (94.6)
Neurologic events	-	1 (3.3)	8 (26.7)	6 (20.0)	15 (50.0)	-	-	10 (33.3)	23 (76.7)
Healthcare-associated infections	-	-	12 (80.0)	3 (20.0)	-	-	-	12 (80.0)	11 (73.3)
Bleedings	-	-	8 (72.7)	2 (18.2)	-	1 (9.1)	-	4 (36.4)	9 (81.8)
Internal events	-	1 (9.1)	8 (72.7)	2 (18.2)	-	-	-	3 (27.3)	10 (90.9)
Allergic reactions	-	-	7 (87.5)	1 (12.5)	-	-	-	1 (12.5)	7 (87.5)
Falls	1 (12.5)	5 (62.5)	2 (25.0)	-	-	-	-	3 (37.5)	6 (75.0)
Pressure ulcers, category I°	-	-	4 (100)	-	-	-	-	4 (100)	3 (75.0)
Total	45 (26.5)	17 (10)	75 (44.1)	17 (10)	15 (8.8)	1 (0.6)	-	110 (64.7)	142 (83.5)

Abbreviations: ^1^ NCC MERP National Coordinating Council for Medication Error Reporting and Prevention, C an error that reached the patient but did not cause harm, D an error that reached the patient and required monitoring or intervention to confirm that it resulted in no harm to the patient, E temporary harm to the patient, F temporary harm to the patient and required initial or prolonged hospitalization or out-patient treatment, G permanent patient harm, H intervention required to sustain life, I patient death.

**Table 5 ijerph-19-02796-t005:** Appended stroke module with frequency of trigger occurrence and positive predictive value.

Stroke Trigger Module	No ^1^ of Triggers Detected in Primary Review	No of Triggers Related to AE ^2^	PPV ^3^ for Triggers Related to AE, %	No of Triggers Related to No-Harm Incidents	PPV of Triggers Related to No-Harm Incidents, %	PPV Total, %
EVT ^4^	20	6	30.0	1	5.0	35.0
Neurological decline of the GCS ^5^ ≥ 4 from the intimal score	4	1	25.0	0	0	25.0
Systemic administration of rtPA ^6^	27	5	18.5	0	0	18.5
Neurological decline of the NIHSS ^7^ ≥ 4 from the initial score	19	2	10.5	0	0	10.5
Thrombin time 1 ≤ 120 s and thrombin time 2 ≤ 4–8 s while under therapeutic heparin within 24 h from onset	32	0	0	0	0	0
Systolic blood pressure above 185 mmHg ^8^ during rtPA administration or in accordance with the neuroradiological report	3	0	0	0	0	0
Computer tomography brain scan ≤ 12 h after rtPA administration	1	0	0	0	0	0
Administration of coagulation factors	0	0	0	0	0	0
Total	106	14	13.2	1	0.9	14.2

Abbreviations: ^1^ No number, ^2^ AE adverse event, ^3^ PPV positive predictive value, ^4^ EVT endovascular treatment, ^5^ GCS Glasgow Coma Scale, ^6^ rtPA recombinant tissue plasminogen activator, ^7^ NIHSS National Institution of Health Stroke Scale, ^8^ mmHg millimeters of mercury.

## Data Availability

The data presented in this study are available on request from the corresponding author.

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
