# Peer review of "Occurrence of No-Harm Incidents and Adverse Events in Hospitalized Patients with Ischemic Stroke or TIA: A Cohort Study Using Trigger Tool Methodology"

_ijerph, 2022, doi:10.3390/ijerph19052796_

Round 1

Reviewer 1 Report

In this article, electronic records from 150 patient admissions for ischemic stroke or TIA were reviewed using the Global Trigger Tool (GTT). The authors found 108 adverse events (AE) and 62 no-harm incidents (NHI), most of them (64.7%) were preventable. While the study is interesting and instructive, I have several questions:

  1. Reporting NHIs in a GTT study is quite unusual. In the introduction section, the authors state that “the information on NHIs provides an additional perspective on patient safety during care services”. According to Table 3, the event types are quite different between NHIs and AEs. Of note, all the medication events were NHIs, which means they were 100% preventable but did not harm the patients. Do these results provide any specific perspective for the care services of ischemic stroke or TIA patients? Do the authors recommend including NHIs for all GTT studies or just for the present study? The authors should discuss it.
  2. Is there any difference between the patietns who experienced AEs and those who did not? Are these two groups statistically different in length of stay (LOS), treatment, NHISS on admission, and mRS after 3 months? The authors may provide this information in Table 1 and/or in the result section.
  3. The authors developed a stroke module with 8 stroke specific triggers but stated (in the discussion section) that the events identified by the stroke specific triggers could be covered by the trigger “any other complication”. In the supplemental material (Table S1), the number of triggers related to AEs for “any procedure complications” is 13, while the number of triggers related to AEs for the entire stroke module is also 13 (Table 5). Does it mean that the latter is totally covered by the former? If it is true, the authors should specify this finding in the result section to make their point clear (that the stroke module did not make additional value). If it is not true, can the authors provide data about how many events would have been missed without the stroke module?
  4. In the supplemental material, the trigger “other” was used 226 times during the primary review, and the PPV was about 50%. According to the IHI GTT guide, the definition of trigger “other” of the care module is “a (potential) event uncovered that does not fit a trigger”. Can the authors elaborate the details of these AEs or NHIs not fitting a trigger? What is about the other half of the potential events (trigger “other”) that were discovered but not related to AEs or NHIs? Is it common for the trigger “other” to be ticked so frequently? If it is not, why?
  5. Although the IHI GTT guide suggests a 20-minute limit for primary review, this study did not set a limit for screening. However, according to the supplemental material, the screening time was generally within 20 minutes (First reviewer median 13’, IQR 8’31”, 21’31”; Second reviewer: median 10’, IQR 6’2”, 16’52”), and the interrater reliability was still good. Does it mean that the 20-minute rule is still applicable for ischemic stroke patients?
  6. The “EVT” trigger was ticked 20 times (Table 5) while only 10 patients received EVT (EVT, n = 6, rtPA + EVT, n = 4) (Table 1). The “systemic administration of rtPA” trigger was ticked 27 times while 26 patients received rtPA (systemic rtPA, n = 22, rtPA + EVT, n = 4). Please explain the discrepancy between the number of ticking and the number of interventions.
  7. At the bottom of Table 3, the total % of NHI should be 36.5% (62/170), not 100%, and the total% of AEs should be 63.5% (108/170), not 100%.
  8. How many patients were admitted to the stroke center during the study period (April 1, 2017 to March 31, 2018)? How many patients met the inclusion criteria and hence were eligible for this study? How many of them had ischemic stroke and TIA, respectively? How many patients were excluded because of hemorrhagic stroke? Please provide this information in the article.

Reviewer 2 Report

This study is a descriptive study which is easy to understand, and a novelty of the study was relatively clear. Just some points need to be clarified.

  1. (Introduction) Although it is written that there is only one study that investigated no-harm incidence and AEs among stroke patient suing GTT, are there any studies using other methods for investigating AEs among stroke? If “no-harm incidence and AEs” are concepts that are particularly specific to GTT, it is not surprising that there are few previous studies.
  2.  (Introduction) Why did you focus on patients with ischemic stroke and exclude patients with hemorrhagic stroke in this study ?
  3. (Methods. line 95) I could not understand the meaning of the sentence. Could you add an explanation ?
  4. (Discussion, line 259) Where is the explanation of the modification of GTT in the Methods? In addition, is it OK to modify the GTT?
  5. (Conclusion) Where is the explanation of the modification of GTT in the Methods? In addition, is it OK to modify the GTT?
